# Changes in locus wide repression underlie the evolution of *Drosophila* abdominal pigmentation

**Iván D. Méndez-González**[1], **Thomas M. Williams**[2], **Mark Rebeiz**[1]*

**1** Department of Biological Sciences, University of Pittsburgh, Pittsburgh, Pennsylvania, United States of America, **2** Department of Biology, University of Dayton, Dayton, Ohio, United States of America

* rebeiz@pitt.com

## Abstract

Changes in gene regulation represent an important path to generate developmental differences affecting anatomical traits. Interspecific divergence in gene expression often results from changes in transcription-stimulating enhancer elements. While gene repression is crucial for precise spatiotemporal expression patterns, the relative contribution of repressive transcriptional silencers to regulatory evolution remains to be addressed. Here, we show that the *Drosophila* pigmentation gene *ebony* has mainly evolved through changes in the spatial domains of silencers patterning its abdominal expression. By precisely editing the endogenous *ebony* locus of *D. melanogaster*, we demonstrate the requirement of two redundant abdominal enhancers and three silencers that repress the redundant enhancers in a patterned manner. We observe a role for changes in these silencers in every case of *ebony* evolution observed to date. Our findings suggest that negative regulation by silencers likely has an under-appreciated role in gene regulatory evolution.

**Data Availability Statement:** The authors confirm that all data underlying the findings are fully available without restriction. All relevant data are within the paper and its Supporting information files.

## Author summary

A central concept in the evo-devo field is that morphological evolution often involves changes in gene regulation. For a variety of reasons, most of the work has focused on the function of enhancers which activate gene expression in discrete patterns. However, this is because enhancers are easier to identify and characterize than other *cis*-regulatory elements that depend on the activity of multiple elements through long-range interactions. Here, we examined the role of interacting *cis*-regulatory elements in the regulation and evolution of the pigmentation gene *ebony* in *Drosophila*. We showed that in *D. melanogaster*, *ebony* abdominal expression is regulated by five *cis*-regulatory elements. Surprisingly, we found that evolutionary changes to the *ebony* silencers are sufficient to explain inter specific differences in expression patterns. Our results highlight the importance of silencers in the evolution of *ebony* regulation and point to a broader possible impact of silencers in the evolution of gene expression that may be prevalent but yet unnoticed.

**Funding:** This work was supported by funding from the NIH (R35GM14196 to MR) and the NSF (IOS2211833 to TMW). The funders had no role in study design, data collection and analysis, decision to publish, or preparation of the manuscript.

**Competing interests:** The authors have declared that no competing interests exist.

## Introduction

Morphological evolution largely depends on changes in the expression of key developmental genes and their downstream target genes [1, 2]. At the core of this process are *cis*-regulatory sequences known as enhancers, which are responsible for activating transcription in a specific spatiotemporal pattern [3]. Enhancers have been the focus of gene regulatory studies for several good reasons: they are typically discovered through reporter assays that test sufficiency and are most commonly found when a regulatory region is dissected. Although enhancers provide a good approximation of gene expression patterns, oftentimes they do not fully recapitulate the endogenous gene expression [4]. This highlights the importance of other types of regulatory sequences, including boundary elements [5], Polycomb response elements [6], silencers [7], and sequences that lie at the outskirts of minimally defined enhancers [8], which interact with enhancers to accomplish precise spatiotemporal patterns of expression. Hence, a key task to understand the evolution of gene regulation is to pinpoint the influence of regulatory elements beyond enhancers, and every example provides key precedents that expand our conception of possible mechanisms.

Transcriptional repression has long been appreciated as an integral component of gene regulation [9–11]. Transcriptional silencers are *cis*-regulatory sequences that repress transcription from otherwise active promoters [12]. Recent evidence hints at the widespread prevalence of silencers in animal genomes [13–15]. However, the difficulty of genomically identifying and functionally characterizing these regulatory elements [12] has limited our ability to test whether the modification of silencer function could be a general mechanism of morphological evolution (but see [16]). Many mechanisms have been proposed for silencer function, from promoter-proximal mechanisms involving histone methylation, to distal elements capable of repressing at long ranges [7]. Because of the long-range character of these elements, they are very difficult to identify by traditional reporter tests of sufficiency. Moreover, since these regulatory elements are able to completely shut down transcription in a patterned manner, they may represent a substantial source of phenotypically relevant genetic variation.

*Drosophila* melanic pigmentation represents a rapidly evolving trait that has provided many insights into regulatory and morphological evolution [17]. In particular, the *ebony* gene presents an intriguing model for understanding regulatory evolution because of its negative regulatory elements. *ebony* encodes an enzyme that decreases the production of black melanin pigments [18]. In *D. melanogaster* males, *ebony* expression anticorrelates with the melanic pigments that adorn the adult abdomen, as it is restricted from the posterior part of the abdominal segments A2-A4 and down-regulated in entire A5 and A6 segments [19]. This expression pattern is controlled by multiple regulatory elements (Fig 1A) [19, 20]. An upstream enhancer drives expression in the entire abdomen (hereafter referred as *eAct*) [19]. A promoter-proximal silencer represses ebony in the A5 and A6 segments of males (hereafter referred as *eMS*) [19]. And an intronic silencer represses ebony in the most posterior region of each segment (hereafter referred as *eSS*) [19]. Recently, it was found that *eAct* also functions as a dorsal midline silencer and that it controls ebony abdominal expression together with yet unidentified redundant enhancers [20].

*ebony* has been implicated repeatedly in the evolution of *Drosophila* pigmentation, and in all cases, *cis*-regulatory rather than coding changes were involved [16, 19, 21–23]. For instance, it was shown that the function of *eMS* is conserved in *D. prostipennis* and *D. yakuba* [21, 23], but not in *D. serrata* nor *D. santomea*, two species that secondarily lost male A5 and A6 melanic pigmentation [16, 23]. Relatedly, this silencer's function was found to be polymorphic in *D. auraria* [16]. These findings are illustrative examples that morphological evolution can

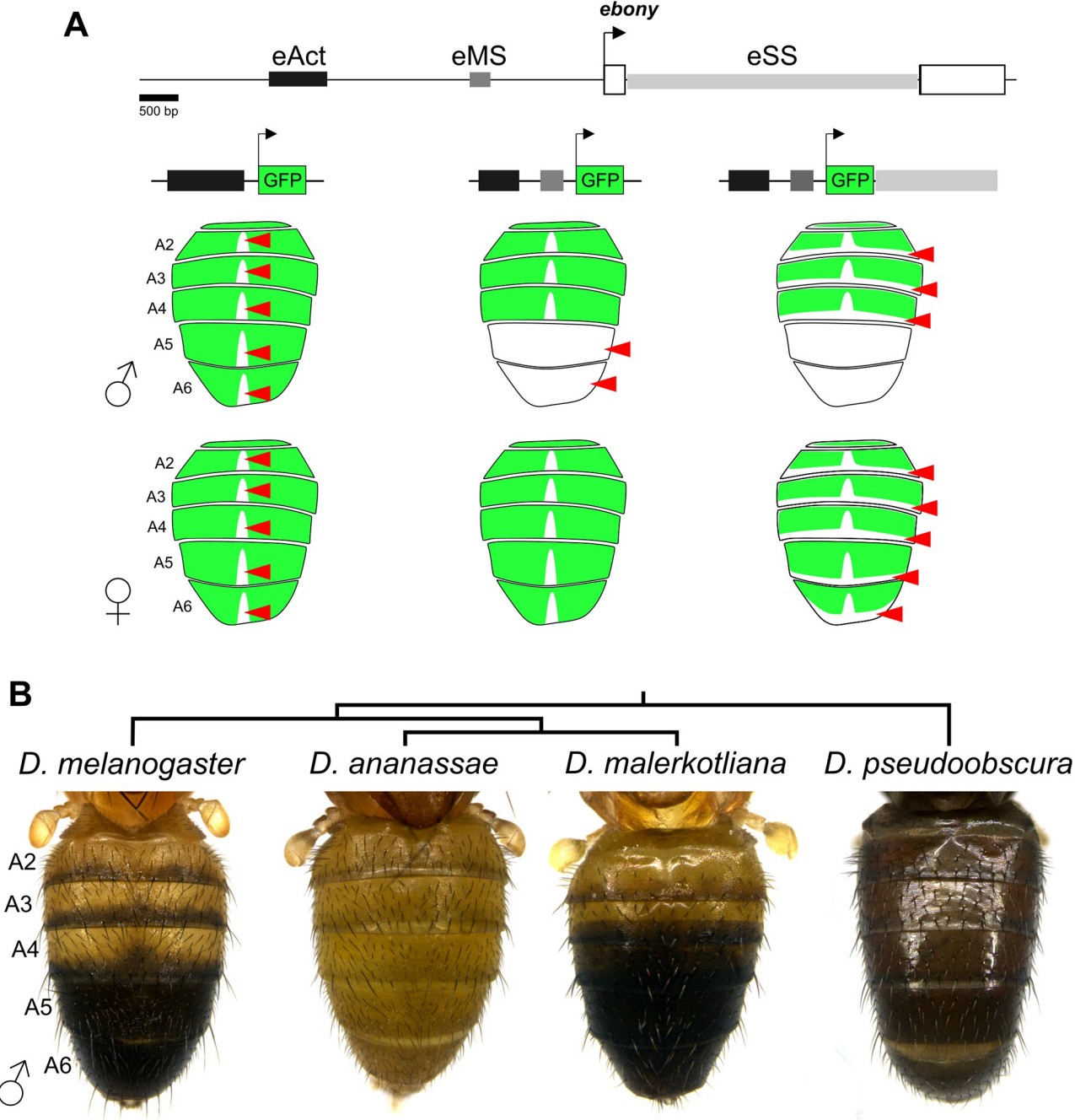

**Fig 1. *ebony* abdominal expression is patterned by multiple regulatory elements.** A: Gene map of the *ebony* locus showing the first two *ebony* exons (white boxes) and the location of known enhancers and transcriptional silencers active in the abdomen of *D. melanogaster*. The cartoons below represent the GFP reporter expression of the upstream enhancer alone and in combination with the two silencers in males (top) and females (bottom). B: Phylogeny showing the abdominal pigmentation of males from different *Drosophila* species.

evolve via silencer inactivation to increase gene expression. The diversity of melanic pigmentation patterns (Fig 1B) that correlate with *ebony* abdominal expression [22, 24] presents an opportune system in which to investigate how regulatory evolution might recurrently proceed in the context of a complex regulatory architecture.

Here, we investigated the *cis*-regulatory evolution of *ebony* in *D. melanogaster* and relatives displaying a range of pigmentation phenotypes (Fig 1B). We found that changes in the function of silencers, rather than enhancers, have contributed to the most salient differences in *ebony* expression among *Drosophila* species with divergent melanic pigmentation. We identified a novel silencer that seemingly evolved within an abdominal enhancer, functionally equivalent silencers with different genomic locations, and spatial expansions in the domain of a silencer's function. Altogether, these data illustrate multiple manners in which differential negative regulation resulting from changes in the function of transcriptional silencers can contribute to phenotypic diversity.

## Materials and methods

### 0.1 *Drosophila* strains and culture conditions

Fly stocks were reared using standard culture conditions. Wild type species used in this study were obtained from the University of California, San Diego *Drosophila* Stock Center (now known as The National *Drosophila* Species Stock Center at Cornell University) (*Drosophila ananassae* #14024–0371.13, *Drosophila malerkotliana* #14024–0391.00, *Drosophila pseudoobscura* #14011–0121.87). The following were obtained from the Bloomington *Drosophila* stock center: *nos-Cas9 (attP40)* (#78781), *cre(III)* (#1501), double balancer (#3703), and *ϕ C31(X)* (#34772). A *D. melanogaster yellow white* (*yw*) strain that was isogenized for eight generations and was used to normalize the backgrounds of GFP reporter transgenes. The line used as WT was created by crossing the *yw* strain with the double balancer line and was used to compare with CRISPR-*Cas9* engineered lines.

### 0.2 CRISPR-Cas9 genome editing

**0.2.1 Design of single guide RNAs (sgRNAS).** To avoid possible off-target effects, sgRNAs were designed using the CRISPR Optimal Target Finder (http://targetfinder.flycrispr.neuro.brown.edu/) and synthesized in vitro. Briefly, 20 nt target-specific primers were designed containing the T7 promoter sequence (upstream) and an overlap with the sgRNA scaffold (downstream). Each target-specific primer was combined with three primers for an overlap extension PCR (0.4 mM each) to generate a 130 bp DNA template. After purification, the template was used for in vitro transcription using EnGen sgRNA synthesis Kit (NEB), and the reaction was cleaned up using the MEGACLEAR Transcription Clean-Up KIT (Thermo). Primer sequences are listed in S1 Table.

**0.2.2 Donor vectors for homologous directed repair.** Homology arms (1.5–2 kb each) were amplified from the *D. melanogaster* strain to be injected and inserted into plasmids containing fluorescent eye markers using NEBuilder Hi-Fi DNA assembly (NEB). Primer sequences and donor plasmids are listed in S1 Table.

**0.2.3 *Drosophila* microinjections.** CRISPR-*Cas9* injections were performed in house following standard protocols (http://gompel.org/methods). All concentrations are given as final values in the injection mix. For the *ebony* loss of function strain, we injected a mix containing a sgRNA targeting the first exon (100 ng/μl), and the plasmids *pCRISPaint-sfGFP-3xP3-RFP* (Addgene 127566) and *pCFD5-frame_selector_0,1,2* (Addgene 131152; 400 ng/μl each) into nos-*Cas9* (attp40). This resulted in the insertion of *pCRISPaint-sfGFP-3xP3-RFP* in the first exon via non-homologous end joining, leading to a loss of function allele [25].

For deletions of the *ebony* non-coding regions, we injected a mix containing the donor vector (500 ng/μl) and one to three sgRNAs flanking each side of the targeted region (100 ng/μl each). For eActΔ, eMaleSilΔ, and eActB + In.4Δ, and eUps + In.4Δ, the EnGen Spy Cas9 NLS (NEB) was added to the mix. eActBΔ, eUpsΔ, and eIn.4Δ were obtained by injecting into the

nos-Cas9(attP40) strain (BDSC 78781). The progeny of each injected fertile individual was screened for dsRed, RFP or GFP fluorescence in the eyes and the correct genomic incorporation of this marker was confirmed by PCR followed by sequencing (see S1 Table for primer sequences). Transformant individuals were crossed with a *yw* strain to remove the nos-*Cas9* transgene, and with a third chromosome balancer strain (BDSC 3703) to produce a stable homozygous line.

### 0.3 Pigmentation quantification

Images of abdomens throughout the manuscript are representative images acquired by mounting 7- to 8-day-old adults on double sticky tape and imaging with a Leica M205C stereo microscope. To quantify the abdominal pigmentation, 10 cuticle preparations from adult flies were used for each genotype and sex. Briefly, flies were aged to 7–8 days old and stored for 2–3 days in ethanol 75% before dissection. Abdominal cuticles were cut through the dorsal midline, which is therefore not visible in the preparations. After dissection, cuticles were mounted in PVA mounting medium (Bioquip). Cuticle preparations were imaged using a Leica M205C Stereo Microscope with a DFC425C camera. Image analysis was performed in ImageJ [26]. Before image analysis, files were blinded using the ImageJ extension LabCode. To measure the percentage of darkness, the anterior portion of segments A4, A5, and A6 was selected to obtain the grayscale darkness value that lies on a 0–255 scale. The percentage of darkness was calculated as: (255-grayscale darkness)/255 × 100 [19]. To measure the A4 stripe thickness, the length of the stripe was measured and divided by the total length of the segment. Boxplots were created using the R [27] packages ggplot2 [28] and ggpubr [29].

### 0.4 *in-situ* hybridization

To obtain information about the spatiotemporal pattern of *ebony* expression, *in-situ* hybridization was performed as described in [23] with small modifications. In brief, flies were collected no more than 30 minutes after eclosion, abdomens were heated for 37 seconds to relax abdominal body muscles, dissected in cold PBS, and fixed in PBS containing 4% paraformaldehyde (E.M.S. Scientific) and 0.1% Triton X-100. PCR was performed to generate an RNA probe template that had a T7 promoter appended through primer design. Primer sequences are listed in S1 Table. Digoxigenin-labeled probes were generated using a 10X Dig labeling mix (Roche Diagnostics) and T7 RNA polymerase (Promega). Dissected samples were probed using an in-situ hybridization robot (Intavis).

### 0.5 GFP transgenic reporters

*ebony* non-coding regions from different species were amplified via PCR and cloned into the S3AG vector using NEBuilder Hi-Fi DNA assembly (NEB). Primer sequences are listed in S1 Table. Information about the source genome sequences for these constructs is provided in S2 Table. *D. melanogaster* transformant lines were generated by ΦC31 mediated site specific recombination into the 51D insertion site on the second chromosome. Injections were performed by BestGene Inc. For all reporters, samples were aged 24h after eclosion and mounted in halocarbon oil 700 (SIGMA). Images were taken using an Olympus Fluoview 1000 confocal microscope. Samples were imaged with standard settings in which the brightest samples were not saturated. To quantify the silencing activity in A5 and A6, the percentage of GFP expression in A5 and A6 compared to A4 was calculated. For this, the pixel intensity of a squared region measured in the anterior part of A5 and A6 was divided by the value measured in A4 (a segment in which no silencing activity was expected). Hence, a value closer to 1 would indicate no repression, while values closer to 0 suggest high silencing activity. To compare A5 and

stripe silencing in the *D. malerkotliana eUps+IN* and *eUps+In.4* reporters, the pixel intensity of the anterior and posterior part of A4 as well as the anterior part of A5 was measured. The stripe silencing activity was calculated as the intensity of the posterior part of A4 divided by the intensity of the posterior part of A4. The A5 silencing activity was calculated as the intensity of the A5 segment divided by the intensity of the A4 segment. Pixel intensity for GFP expression was quantified using ImageJ [26].

## Results

### 0.6 Redundant enhancers contribute to *ebony* abdominal expression in *D. melanogaster*

A recent study found that deleting the main abdominal enhancer (*eAct*) does not notably affect *ebony* expression, suggesting the presence of redundant enhancers [20]. However, the number and location of such enhancers has not been determined. We used CRISPR-*Cas9* to create a series of deletions aiming to identify the redundant enhancer(s) (Fig 2A). *ebony* null mutants develop a darker pigmentation compared to wild type controls (WT, Fig 2B–2C'), setting the expectation that flies will become *ebony*-like once all redundant enhancers are removed. Deletion of *eAct* did not increase the A4 percentage of darkness (Fig 2D, 2J and 2K), confirming previous results [20]. However, it resulted in the loss of the dark midline stripe (see below). We wondered whether important sequences that maintain WT levels of *ebony* expression reside outside of the deleted region. To test this, we deleted an expanded region centered on *eActΔ* (*eActBΔ*), and the entire upstream region (*eUpsΔ*). Both deletions resulted in slightly darker flies compared to WT, although still considerably lighter than *ebony* null mutants (Fig 2E, 2F, 2J and 2K).

Even though these deletions only had a mild effect in the adult pigmentation, we wondered if they had any effect on *ebony* expression. We opted for spatial measurements of *ebony* mRNA in the abdomen of flies at the eclosion stage using *in situ* hybridization. While all deletion backgrounds showed WT levels of expression, deletions overlapping the *eAct* region resulted in *ebony* de-repression along the dorsal midline (S1 Fig). These expression patterns correlate with the adult pigmentation of these lines in which the dorsal midline melanic stripe is erased (S1B–S1I' Fig) and confirm the function of this region as a silencer [20]. These results suggest that redundant enhancer(s) located outside the *ebony* upstream region work together with the element in the *eActB* region to ensure WT levels of expression in the abdomen.

To identify the redundant enhancer(s), we focused on a candidate region located within the first *ebony* intron (*eIn.4*, Fig 2A). This region was identified as a putative abdominal enhancer in *Drosophila* species from the *ananassae* subgroup [22]. Importantly this candidate region does not overlap with the intronic stripe silencer *eSS* (see below). We reasoned that a possible redundant enhancer could be identified by deleting this region in the *eActBΔ* or *eUpsΔ* backgrounds. The deletion of the candidate region alone (*eIn.4Δ*) did not affect the pigmentation (Fig 2G and 2G'). However, both double deletions, *eActB+In.4Δ* and *eUps+In.4Δ*, resulted in much darker pigmentation compared to the single deletions and approaching to the pigmentation of *ebony* mutants (Fig 2H–2K). Thus, *eIn.4* functions as a partially redundant enhancer working together with *eActB* to drive robust ebony expression in the abdomen.

Although we focused on the abdominal pigmentation, we noticed that other tissues including the head, thorax, legs, and halteres, of *eUps+In.4Δ* had a darker pigmentation compared to WT (S2 Fig). Enhancers responsible for *ebony* expression in these tissues have been mapped to the upstream region [19]. However, the pigmentation of these tissues in *eActBΔ* and *UpsΔ* appears WT (S2 Fig). Thus, *eIn.4* represents a redundant enhancer that is active in multiple adult tissues. Altogether, these experiments revealed a complex mechanism for *ebony*

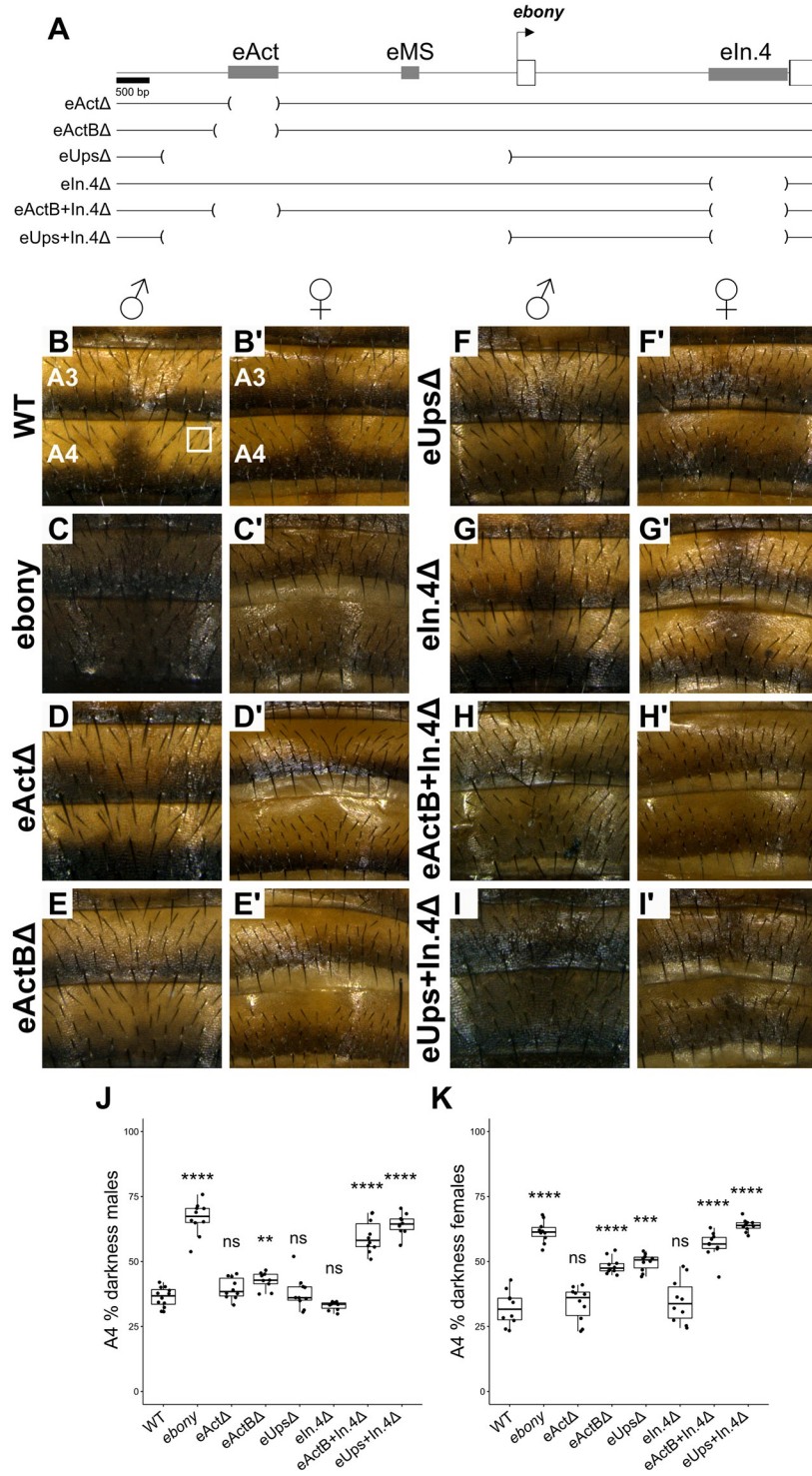

**Fig 2. *ebony* abdominal expression is controlled by redundant enhancers.** A: Gene map of the *ebony* locus showing the location of the first two *ebony* exons (white boxes) and of the deletions created to identify redundant enhancers. B-I': A3 and A4 pigmentation of WT, *ebony* null mutants, and deletion lines males and females. The white square in B shows the region used to measure the percentage of darkness. J-K: Quantification of the A4 percentage of darkness of males and females. Significant differences are shown compared to WT. (Student's t test, ns = not significant, $^*p < 0.5$, $^{**}p < 0.05$, $^{***}p < 0.005$, $^{****}p < 0.0005$).

regulation in which upstream tissue-specific enhancers collaborate with an intronic epidermal redundant enhancer to ensure robust expression in the adult cuticle.

## 0.7 *ebony* abdominal silencers are active in specific spatial domains

Gene reporter analysis suggests that *ebony* repression in the male A5 and A6 segments is mediated by a silencer referred to as *eMS* [19]. To confirm the function of *eMS* in its endogenous context, we created a deletion targeting this region (Fig 3A). Surprisingly, the A5-A6 pigmentation was not affected in *eMSΔ* (Fig 3B, 3C, 3H and 3I). To test whether this deletion could modify the spatial expression of *ebony mRNA* we performed in situ hybridization in this line and compared it to WT. We observed a qualitative increase in *ebony mRNA* in the A5-A6 segments of eMSΔ (Fig 3E and 3F). Hence, these experiments confirm that *eMS* is necessary to repress ebony in the A5 and A6 male segments.

*ebony* expression is also repressed in the area where the posterior melanic stripes develop by an intronic silencer referred as *eSS* [19]. We narrowed down the exact location of this silencer using nuclear-localized Green Fluorescent Protein (or GFP) reporter constructs containing fragments of the *ebony* first intron. A region of ca. 1.5 kb located downstream of the *ebony* promoter (eUps+In.1) showed low GFP expression in the stripe area (S3 Fig). The endogenous deletion of this region resulted in *ebony* de-repression in the stripe area and thinner melanic stripes compared to the WT (S4 Fig), confirming that this region is the *eSS*. Together, these experiments show that the silencers *eMS* and *eSS* are necessary and sufficient to repress the *ebony* redundant enhancers in specific spatial domains. Interestingly, the phenotypic effects of *ebony* de-repression seem to be specific to each abdominal region. While *eMSΔ* does not modify the A5-A6 pigmentation, *eActΔ* and *eSSΔ* result in the lack of midline stripe and narrower posterior stripes, respectively. These differences could be explained by the counteracting effects of genes with an opposite function to *ebony*, like *tan* and *yellow* [30, 31] and suggest that these genes may be expressed at different levels across the abdomen.

## 0.8 Changes in the function of silencers drive the evolution of *ebony* expression among *Drosophila* species

To understand how *ebony* expression has evolved, we analyzed its regulation in three additional *Drosophila* species. *ebony* has been identified as a major driver of pigmentation diversity within the *ananassae* species subgroup [22]. Thus, we selected two species from this group with contrasting abdominal pigmentation, *D. ananassae* (non-melanic) and *D. malerkotliana* (A4, A5 and A6 melanic). We also included *D. pseudoobscura*, a completely melanic species which displays very low levels of *ebony* expression [24] (Fig 4A). We created three reporter constructs for each species, containing the region orthologous to the upstream abdominal enhancer (*eAct*), the entire upstream region (*eUps*), and the upstream and first intronic region (*eUps+IN*, Fig 4B). These constructs were tested for GFP activity in the A4-A6 segments of transgenic *D. melanogaster* males 24 hours (h) after eclosion. The relative level of expression in A5 and A6 compared to A4 was used as a proxy for A5-A6 silencing activity.

We found that the activator region of *D. ananassae* drives reporter expression in all abdominal segments (Fig 4C). Qualitatively, this expression pattern did not change when the full upstream region (Fig 4D) or upstream together with the intronic regions were analyzed (Fig 4E). These results suggest that in *D. ananassae*, *ebony* abdominal expression is controlled by an upstream enhancer (Fig 4L).

For *D. malerkotliana*, we found that the activator and the upstream region drive uniform GFP expression in all abdominal segments (Fig 4F and 4G). This reporter activity does not recapitulate *ebony* endogenous expression of *D. malerkotliana*, which is restricted from the

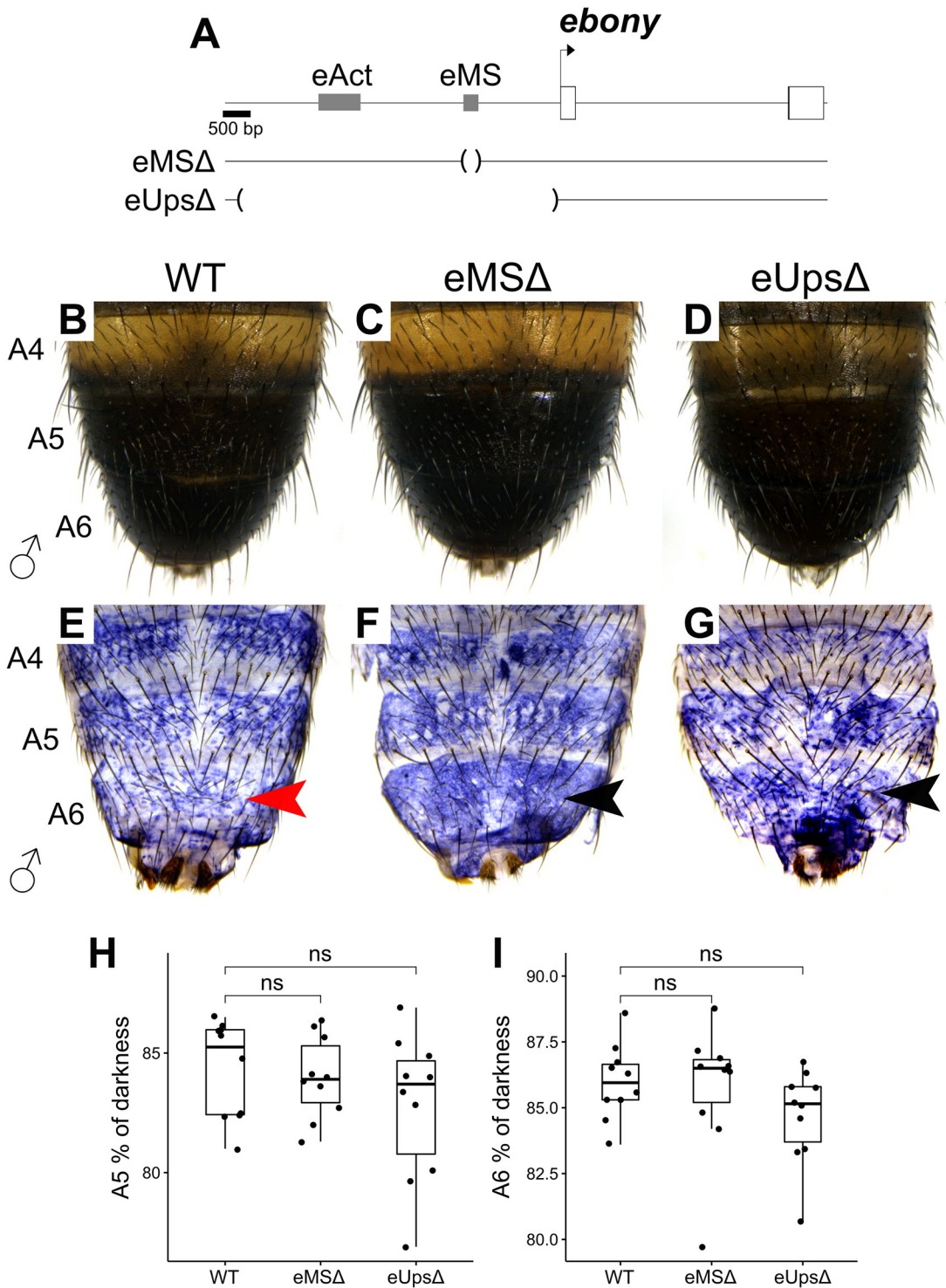

**Fig 3. Necessity of the *ebony* A5/A6 male silencer.** A: Gene map of the *ebony* locus showing the location of the first two *ebony* exons (white boxes) and of the deletion targeting the A5/A6 male silencer (*eMSΔ*). B-D: A4, A5, and A6 pigmentation of WT, *eMSΔ*, and *eUpsΔ* males. E-G: *in-situ* hybridization detecting the spatial expression of *ebony mRNA* in A4, A5 and A6 segments of WT, *eMSΔ*, and *eUpsΔ* males. Red and black arrowheads indicate low and increased levels of *ebony mRNA*, respectively. H-I: Comparison of A5 and A6 darkness between WT, *eMSΔ*, and *eUpsΔ* males. (Student's t test, ns = not significant).

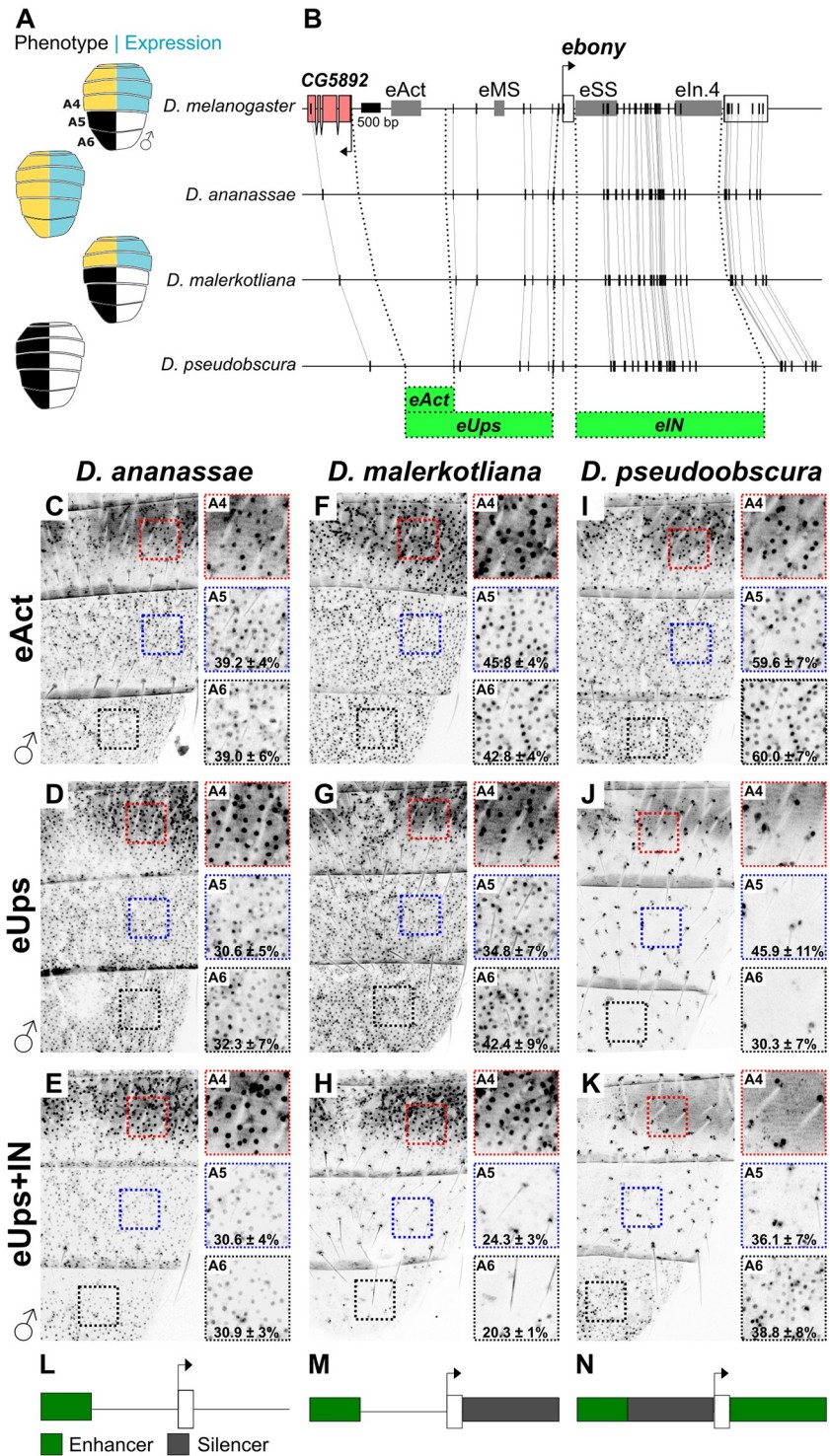

**Fig 4. Changes in the location and function of transcriptional silencers among *Drosophila* species.** A: Cartoons representing the pigmentation phenotype (left) and the *ebony* expression pattern (right, blue color) of *D. melanogaster*, *D. ananassae*, *D. malerkotliana*, and *D. pseudobscura* males. B: Gene map showing the location of the first two *ebony* exons (white boxes) and CREs in *D. melanogaster* (top). Vertical lines indicate syntenic regions, while dashed lines indicate the fragments used to create reporter constructs (green, bottom). C-K: GFP expression patterns of the indicated GFP reporters in the posterior abdominal segments A4-A6 of transgenic *D. melanogaster* males. Insets show magnified regions for A4 (red square), A5 (blue square), and A6 (black square). Numbers show the relative percentage of GFP expression and the standard error of the mean in A5 and A6 compared to A4. All flies were imaged 24 h post

ecolosion. L-N: Inferred *ebony* regulatory architecture showing the approximate location of abdominal enhancers (green) and silencers (gray) for *D. ananassae*, *D. malerkotliana*, and *D. pseudoobscura*, respectively.

A4, A5, and A6 segments (Fig 4B, [22]). However, when the intronic region was included, the expression in A5 and A6 decreased almost by half (A5, 34.8% to 24.3% and A4, 42% to 20%, was silenced, Fig 4H), suggesting the presence of an intronic A5-A6 male-specific silencer. The lack of A4 repression, which is observed in the *ebony* endogenous expression in this species, could result from changes in the trans landscape compared to *D. melanogaster*, or an unidentified A4 silencer. We noticed that the *D. malerkotliana eUps+IN* reporter also repressed GFP expression in the stripe area (Fig 4H). This suggests that this species contains intronic silencer (s) active in both the A5-A6 segments and in the stripe area. We wondered whether the male silencer may be located in an intronic region implicated in the pigmentation differences between *D. malerkotliana* (pigmented) and its sister species *D. malerkotliana pallens* (not pigmented) [22], while the stripe silencer might be orthologous to the *D. melanogaster eSS*. GFP expression of a reporter containing the upstream and the candidate intronic regions (*eUps +In.4*) was repressed in A5-A6, but not in the stripe area (S5 Fig). Thus, the *In.4* region contains the male silencer and might indeed underlie the pigmentation differences between *D. malerkotliana* and its sister species, while the stripe silencer seems to be conserved with respect to that of *D. melanogaster* (S5 Fig). These results suggest that in *D. malerkotliana*, *ebony* abdominal expression is controlled by an upstream enhancer and at least two tissue-specific silencers (Fig 4M).

For *D. pseudoobscura*, we found that the activator region drives GFP expression in A4-A6 segments in a similar pattern to *D. ananassae* and *D. malerkotliana* (Fig 4I). This was surprising considering that the endogenous expression of *ebony* in *D. pseudoobscura* is almost undetectable [24]. However, when the full upstream region was analyzed, we found no GFP expression throughout the abdomen (Fig 4J). This suggests that *D. pseudoobscura* has a functional abdominal enhancer, which is repressed by a silencer located between this enhancer and the *ebony* promoter. When the upstream and intronic regions were analyzed together, we observed GFP expression only in A6 albeit at low levels (Fig 4K). We analyzed the reporter expression of the intronic region alone and found it to be A6 specific (S6 Fig). These data suggest that the low *ebony* abdominal expression of *D. pseudoobscura* [24] results from a silencer that represses *eAct* in all abdominal segments but seems unable to repress the A6 intronic enhancer (Fig 4N).

## 0.9 Evolution of the melanic dorsal midline through the gain of a novel silencer

The melanic stripe that forms along the dorsal midline in *D. melanogaster* (Fig 1B) is regarded as characteristic of species within the subgenus *Sophophora* [32]. However, we have not observed this pigmentation trait in species from the *ananassae* or *montium* subgroups. Given that the formation of the melanic dorsal midline requires *ebony* repression via the silencer activity of *eAct* (S1 Fig) [20], we wondered about the evolution of this silencer function. We found that the *eAct* transgenic reporter of the three species studied here drive robust GFP expression along the dorsal midline (S7 Fig), suggesting that none of these species contain a functional midline silencer. To expand our phylogenetic sample, we analyzed the ebony midline expression and silencer function using published data for *D. prostipennis*, *D. serrata*, *D. auraria*, *D. yakuba*, and *D. santomea* [16, 21, 23]. None of these species showed evidence of

*ebony* midline repression or of a functional midline silencer (S7 Fig). Thus, the silencer function of *eAct* seems to be novel to *D. melanogaster* and may have contributed to the evolution of the melanic dorsal midline.

## Discussion

The importance of silencers for patterning gene expression in metazoans has long been recognized [33]. However, this mode of negative regulation has been difficult to study due to limited examples and heterogeneous mechanisms of action [7, 12]. We showed that multiple silencers are required for patterning spatial and sex-specific *ebony* abdominal expression, and that changes in the function of these silencers have resulted in altered expression patterns contributing to variation in abdominal pigmentation. Interestingly, the ability of *ebony* silencers to antagonize redundant enhancers appears to be case-specific. Below, we reconstruct the evolution of the *ebony* regulatory architecture and discuss how current experimental practices might obscure the significance of silencer evolution in the study of regulatory evolution (Fig 5).

### 0.10 Evolutionary history of a complex regulatory architecture

*D. melanogaster* has evolved a complex assemblage of two enhancers and three tissue-specific silencers required for shaping *ebony* abdominal expression. Comparative analysis of our reporter constructs suggests that each *ebony cis*-regulatory element has a unique evolutionary history. The upstream enhancer (*eAct*) seems to have evolved, at least, in the common ancestor of the *melanogaster-obscura* species groups. However, the dual function of this region as a dorsal midline silencer [20] appears novel to *D. melanogaster*, where it seems to have contributed to the evolution of the melanic dorsal midline. Regarding *eMS*, we propose that the common ancestor of the *melanogaster-obscura* groups possessed a functional upstream silencer, as *D. pseudoobscura* also contains an upstream silencer (which is active in both sexes). After the divergence of these lineages, this silencer acquired a male-specific function specifically in the *melanogaster* group, which coincides with the evolution of male-specific melanic pigmentation

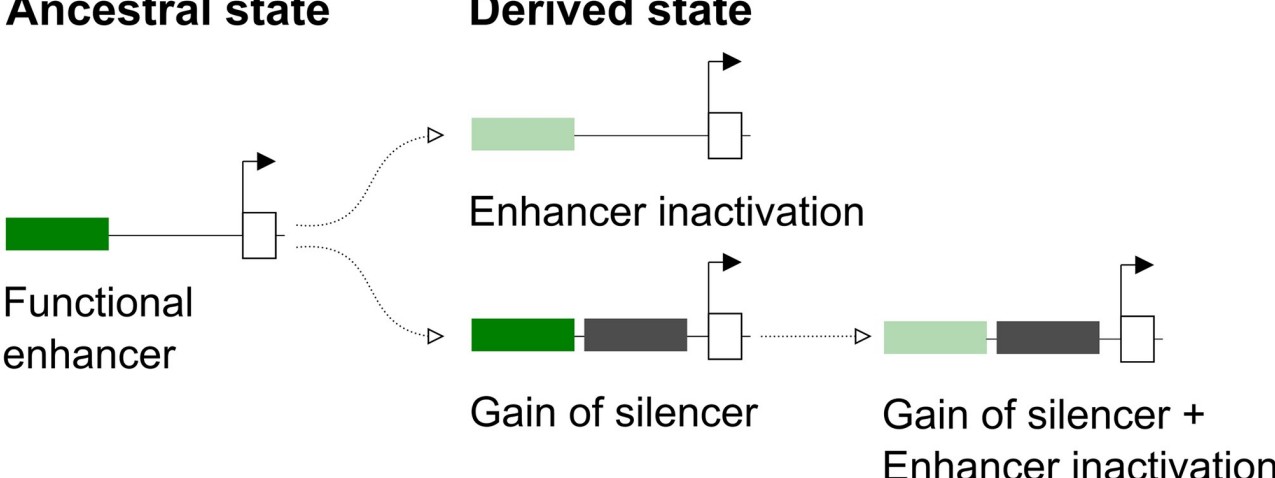

**Fig 5. Different regulatory scenarios resulting in the loss of tissue-specific expression.** In the ancestral state, a gene is regulated by a enhancer (green box). The loss of expression in the derived states may involve the inactivation of its enhancer (top) or the gain of a silencer (bottom, gray box). In the latter, the enhancer may remain functional or eventually become inactive.

[34]. However, the *ananassae* subgroup seems to have gained an intronic male-silencer, while losing the upstream silencer activity. Interestingly, the *D. malerkotliana* male-silencer maps to the same genomic region as the redundant intronic enhancer of *D. melanogaster*. Although challenging, future work involving these intronic regulatory elements might help to elucidate how enhancer logic and silencer logic could interconvert.

## 0.11 Loss of expression by increased negative regulation of a functional enhancer

The characteristic dark pigmentation of *D. pseudoobscura* correlates with low and high *ebony* and *yellow* expression, respectively [24, 30]. Unexpectedly, we found that this species has a functional *ebony* abdominal enhancer that is likely homologous to the *D. melanogaster eAct*. However, a silencer active throughout the abdomen strongly represses this enhancer. Of note, the ubiquitous silencer of *D. pseudoobscura* is not able to repress the A6 intronic enhancer. This provides an important exception to the observed trend that *ebony* silencers are global rather than selective. Silencers appear to comprise multiple functional classes, characterized by distinct associated proteins and interactions with other regulatory elements [7]. Gisselbrecht and colleagues [13] found that embryonic silencers bound by the Snail repressor likely function by preventing nearby enhancers from activating the transcription of target genes. Snail-unbound silencers, on the contrary, seem to loop directly to promoters where they recruit repressive activities. The second class, thus, would result in repression regardless of enhancer redundancy. Investigating the mechanisms of the *ebony* enhancers and silencers may resolve how differences in the mode of silencer action are encoded.

Morphological evolution often results from loss of tissue-specific expression following enhancer inactivation [2, 34, 35]. An extreme example is the evolution of trichome patterns in *D. sechellia*, which involved the parallel inactivation of multiple enhancers of the *shavenbaby* gene [36]. Our results thus provide a distinct counterexample in which the dark pigmentation of *D. pseudoobscura* might have evolved through strong repression of *ebony* while preserving enhancer functionality. These two paths to evolution would appear to differ in the number of required steps, as inactivation of multiple enhancers would likely involve more mutations than changes to a global silencer (Fig 5). Indeed, it has been posited that evolution of repressor sites in individual enhancers may present a shorter path to loss of expression than the loss of multiple activator sites, by vitue of their dominant mode of action [37]. However, it is important to remember that experimental biases towards enhancer studies, as discussed below, may skew our interpretations.

## 0.12 Transcriptional silencers and morphological evolution

Is the trend of silencer evolution at *ebony* an exception? It is our opinion that the *Drosophila* abdomen reflects an opportune system in which to notice repressive mechanisms that may be more prevalent than currently expected. Compared to microscopic tissues with three-dimensional complexity such as the embryo or imaginal disc, the abdomen is a relatively simple two-dimensional canvas upon which even slight deviations of a reporter gene pattern from the endogenous expression pattern can be easily detected. Thus, a gene subject to silencer regulation, such as *ebony* would be easier to detect in this system (Table 1).

The enhancer-centric way that gene regulatory evolution is studied is also skewed to overlook the potential role of silencers. When a difference in gene expression is found between distantly related species, the only way to determine whether those differences are caused by *cis*-regulatory evolution is to find the responsible enhancer(s) and ask whether they have differing activities using gene reporter constructs tested in a common genetic background [38, 39]. If

**Table 1. Summary of *ebony* regulatory changes that affect male-specific pigmentation in *Drosophila*.**

| Species | CRE affected | Phenotypic change | References |
|---|---|---|---|
| *D. melanogaster* | Dorsal midline **silencer** | Gain of melanic midline stripe | This paper |
| *D. santomea* | Male **silencer** | Loss of A5-A6 male melanism | [23] |
| *D. auraria* | Male **silencer** | Decrease of A6 male melanism | [16] |
| *D. ananassae* | Male **silencer** | Loss of A5-A6 male melanism | This paper |
| *D. malerkotliana* | Genomic relocation of the male **silencer** | - | This paper |
| *D. pseudoobscura* | Monomorphic pan-abdominal **silencer** | Gain of monomorphic abdominal melanism | This paper |

Note: the common ancestor of these species is hypothesized to have had A5-A6 male melanism [34], and gains and losses were inferred based on this ancestral state.

the reporter genes recapitulate differences in expression observed within these species, such a result would be consistent with a *cis*-regulatory basis for these evolutionary differences. On the other hand, interspecific differences in enhancer-reporter expression are often attributed to *trans*-regulatory evolution. And yet, it may well be that these differences are actually encoded by *cis*-regulatory changes affecting silencer function. Considering the relative difficulty of finding and testing silencers [7, 12], it stands to reason that these modes of regulatory evolution are likely to be much more common than previously appreciated. Genomic surveys of open chromatin may offer an avenue to identify silencers and other regulatory elements. Indeed, in the butterfly wing, the endogenous deletion of an ATAC-seq peak region was associated with expanded expression, consistent with silencer function [40]. Thus, as the field of evolutionary-developmental biology seeks to further understand the *cis*-regulatory basis for morphological evolution, it will almost certainly have to contend with silencers and other long-distance inter-acting elements as needles in a vast regulatory sequence's haystack.

## Supporting information

**S1 Fig. *ebony* abdominal *mRNA* expression correlates with pigmentation phenotypes.** A: Gene map of the *ebony* locus showing the location of the deletions created to identify redundant enhancers. B-F' *ebony* abdominal *mRNA* expression measured with *in-situ* hybridization in recently eclosed adults for WT, *ebony* null mutants, and deletion lines males and females.
(TIF)

**S2 Fig. The redundant intronic enhancer is active in multiple adult tissues.** A: Gene map of the *ebony* locus showing the location of the deletions created to identify redundant enhancers. Previously identified tissue-specific enhancers are shown on top of the *ebony* upstream region (shaded rectangle). B-G: Pigmentation of different adult tissues in females from the different strains created. Red arrows show tissues, other than the abdomen, with darker pigmentation compared to the WT and more similar to ebony mutants.
(TIF)

**S3 Fig. Identification of the stripe silencer within the first *ebony* intron.** A: Gene map of the *ebony* locus showing the location of the reporter constructs created to identify the stripe silencer within the first intronic region. B-G: GFP expression pattern of the different trans-genic reporters at 24h after eclosion. Blue and red dashed boxes show a magnification of the stripe area in A3 and A4, respectively.
(TIF)

**S4 Fig. Necessity of the *ebony* stripe silencer.** A: Gene map of the *ebony* locus showing the location of the deletion targeting the stripe silencer (*eSSΔ*). B-E: Adult pigmentation of WT and *eSSΔ* males and females. B-E': *In-situ* hybridization detecting *ebony mRNA* in the A4 segment of WT and *eSSΔ* males and females. Red and black arrowheads indicate low and increased levels of *ebony mRNA*, respectively. F: Comparison of the relative thickness of the melanic stripe between WT and *eSSΔ* males and females (Student's t test, *** = $p < 0.0005$).
(TIF)

**S5 Fig. The *ebony* male and stripe silencers of *D. malerkotliana* are located in distinct intronic regions.** A: Gene map showing the reporter constructs created to identify the location of the *D. malerkotliana* male silencer within the first *ebony* intron. B-C: GFP expression pattern of *D. malerkotliana* transgenic reporter eUps+IN and eUps+In.4. Boxed regions show expression in A4 stripe region (red), and A5-A6 segments (blue and black, respectively). D: Inferred location of the *D. malerkotliana* intronic silencer within the first *ebony* intron.
(TIF)

**S6 Fig. Enhancer activity of the *ebony* intronic region from *D. pseudoobscura*.** A: Gene map showing the reporter constructs created for *D. pseudoobscura*. B: GFP expression patterns of *D. pseudoobscura* transgenic reporter eIN.
(TIF)

**S7 Fig. The melanic dorsal midline is novel to *D. melanogaster*.** A: GFP expression patterns of the eAct transgenic reporters in the abdominal segments A4-A5. Insets show magnified regions along the midline for A4 (red square) and A5 (blue square). B: Phylogenetic distribution of the melanic dorsal midline in *Drosophila* species for which the expression and regulation of *ebony* in this area has been studied.
(TIF)

**S1 Table. List of primers used in this study.**
(XLSX)

**S2 Table. Information about the source genome sequences for the GFP reporter constructs.**
(XLSX)

## Acknowledgments

The authors thank the members of the M.R. laboratory for their comments and discussion on the manuscript. We thank the Bloomington Stock Center, the National Drosophila Species Stock center for fly stocks, and Stephanie Day for help cloning reporter lines.

## Author Contributions

**Conceptualization:** Iván D. Méndez-González, Thomas M. Williams, Mark Rebeiz.

**Data curation:** Iván D. Méndez-González.

**Formal analysis:** Iván D. Méndez-González.

**Funding acquisition:** Thomas M. Williams.

**Investigation:** Iván D. Méndez-González.

**Project administration:** Thomas M. Williams, Mark Rebeiz.

**Supervision:** Mark Rebeiz.

**Validation:** Iván D. Méndez-González.

**Visualization:** Iván D. Méndez-González.

**Writing – original draft:** Iván D. Méndez-González.

**Writing – review & editing:** Iván D. Méndez-González, Thomas M. Williams, Mark Rebeiz.

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
