## [Decision Letter · Decision Letter 0]

9 Jan 2023

Dear Dr Rebeiz,

Thank you very much for submitting your Research Article entitled 'Changes in global repression underlie the evolution of Drosophila abdominal pigmentation' to PLOS Genetics.

The manuscript was fully evaluated at the editorial level and by independent peer reviewers. The reviewers appreciated the attention to an important topic but identified some concerns that we ask you address in a revised manuscript.

We therefore ask you to modify the manuscript according to the review recommendations. Your revisions should address the specific points made by each reviewer.

Yours sincerely,

Artyom Kopp

Academic Editor

PLOS Genetics

Kirsten Bomblies

Section Editor

PLOS Genetics

Reviewer's Responses to Questions

**Comments to the Authors:**

Reviewer #1: “Changes in global repression underlie the evolution of Drosophila abdominal pigmentation” by Mendez-Gonzalez and colleagues highlights something that is rather unexplored in evolution and developmental studies, that of negative regulation by silencers. Throughout the manuscript, the authors perform an elegant series of engineered deletions balanced with reporter gene constructs. Overall, I found this to be a great paper and below highlight a few points that could elevate the good work.

1) I would change the title: “global repression” from a genomics perspective could be misconstrued as genome wide. Maybe something about locus wide repression, or dominant repression.

2) Highlight the midline on the abdomen in Fig. 1A

3) Figure 3 E-G, could these be quantified? It would be nice to see how these deletions affect the levels of mRNA. In other papers (Tsai et al., elife 2019), the difference between RNA levels and phenotypes has been proposed to be through buffering. So, this would be great data to highlight and would strengthen the manuscript.

4) Figure 4A, can you color the “Phenotype|Expression”

5) I would recommend removing the svb from the figure and just save it for the discussion (or a future review), it drops abruptly and distracts a bit from the overall paper and can easily be discussed

6) It has long been discussed by developmental biologists (for example by Eric Davidson) that repression could be “dominant” which would, as the authors note, facilitate rapid evolution. I would recommend fleshing this out in the discussion because it’s a great point!

7) The authors could mention some of the genomics papers that explore insulators (see Barak Cohen's new preprint)

In sum, this was a great manuscript exploring an underappreciated role of negative regulation by silencing.

Reviewer #2: A major goal of evolutionary biology is to identify molecular genetic mechanisms that drive phenotypical differences between different species. In this study, Méndez-González and colleagues investigated the evolution of cis-regulatory elements (CREs) controlling ebony and abdomen pigmentation in the fly. The ebony locus was previously implicated in the evolution of abdominal pigmentation in flies by the last author (Rebeiz 2009 Science), and the present work represents a follow-up study that substantially improves our understanding of the complex cis-regulatory architecture of this locus which involves multiple independent enhancers and silencers elements. The authors further show that changes in silencers contribute to morphological change which is the main novelty of the paper. This work is conceptually novel because most previous work on the role of non-coding DNA in evolution focused on changes in enhancers (loss or gain of function) and ignored silencer regions.

Specifically using in vivo CREs knockout experiments in D. melanogaster with observable direct phenotypic readout (pigmentation darkness of abdominal cuticula), the authors first functionally dissected regulatory regions upstream and downstream ebony promoter. These experiments identify a previously uncharacterized intronic abdomen enhancer of ebony that functions redundantly with a known upstream abdomen enhancer. Deletion of regions containing silencer regions confirms that these regions (previously characterized in transgenic reporter experiments) are required for silencing their endogenous target gene (ebony). Using interspecies transgenic reporter experiments with ebony regulatory regions from three other drosophila species the authors show that the evolution of these silencer elements underlies phenotypic differences between differently pigmented fly species. This finding – that silencer divergence underlies morphological evolution - represents a significant and novel contribution to the field. Before publication, this work will benefit from more rigorous analyses of their knockout experiments and the inclusion of motif analyses, among other comments.

Major comments

1. The authors show that both silencers (eMS and eSS) are sufficient to repress ebony expression in their reporter assay. Surprisingly, however, only the deletion of eSS is sufficient to alter the pigmentation phenotype. How do the authors reconcile this discrepancy? The authors cite that for eMS it could be due to the compensatory expression of yellow and tan genes, but presumably these genes are also expressed in the stripe areas. This should be further discussed. Also the data showing the effects of eSS deletion should be in the main figure 2 together with eMS deletion data.

2. The authors make claims that deletions of silencers, for instance, increase ebony expression, as measured by ISH, but there is no formal quantification supporting this. While the images are helpful, the authors should formally quantify these differences and whether these differences are significant between genotypes to make such conclusions.

3. The scale/units for pigmentation darkness and stripe thickness need to be clarified. For example, in Fig 2, it says ‘relative darkness,’ but there is no description of how this was normalized to WT levels.

4. How do these putative silencers repress ebony expression on a mechanistic level? This remains largely unanswered. It is surprising that the authors don’t discuss this. While looking at mechanistic details of silencing such as histone modifications (deacetylation or depositing repressor marks) is beyond this paper’s scope, the authors could at least consider examining TF motifs among the different silencer sequences and checking if there are gains/losses of putative repressors between different species.

5. The authors show a loss of GFP reporter expression in the presence of the eMS silencer element from D. pseuodobscura. Though beyond the scope of this paper, in my opinion, the most exciting experiment would be to test whether this D. pseuodobscura eMS element is sufficient to repress ebony expression when swapped into the native D. melanogaster locus. To be clear, I’m not asking for the authors to do this, but if they already have this data, this manuscript would benefit from its inclusion.

6. The authors cite relevant work from Gisselbrecht et al. (ref 13) that demonstrated silencers might also function as enhancers in other cellular contexts. Is there any evidence supporting this for the silencers tested in this study? Do they work as enhancers in different tissues or at diferent timepoints? Perhaps it could be examined if there are overlapping characterized Gal4 lines from Janelia or Vienna collection.

Minor comments

1. Is Fig 4C-K only males? This is explicitly outlined in other figures, but not this one. Please include.

2. The use of different names for the same enhancer/silencer element (i.e., eMS and A5/A6 silencer) within the text and figures makes it very difficult to follow. The authors should be consistent in their regulatory element nomenclature.

3. Likewise, one deletion is named IN.4 and another In.4 for different enhancers. This is confusing and the authors should change these names to be more intuitive for the reader.

Reviewer #3: See separate attachment with formatted review.

Reviewer #4: This is an interesting study and a well-written manuscript presenting experimental data, appropriately collected and analysed, that significantly contributes to the mechanisms of transcriptional regulation and its putative roles in morphological evolution.

There are no major flaws to report or fundamental new experiments to carry out but there is room for improving the manuscript. Below is provided a list of general and particular aspects that should be clarified and suggestions that might improve the understanding of the data and of its conclusions.

General comments:

Not enough detail in the methods that complicates the interpretation of the results. Please do have an overall revision with an eye to missing information necessary for the untrained reader. Some examples, below.

1) There is not sufficient detail regarding the quantification methods applied for the three different cases (pigmentation, in situ, and GFP reporters). For instance, it is not obvious from the figures that there was ‘higher ebony mRNA expression compared to WT as measured by in situ hybridization’ (Fig 3D-E), (lines 201-202).

2) Were these differences quantified, as done for darkness (panels H-I)?

3) In general, it would be nice to see some of the quantitative measures done for the GFP reporter experiments (as described in lines 143-149), especially given that some of the differences mentioned are not so obvious from the chosen images (for instance in Fig 4).

4) The authors could also be more concrete on how the define the line that separates the pigmentated from the non-pigmented part of the segment. This is important for the overall quantification analyses, as it is, more specifically, for the same of Fig S4 (panel F) where the relative thickness is presented.

5) It is also not clear how the authors define homology between segments in the analyses of the different species (contained in Fig. 4 A-B). Is there synteny between all species that includes the upstream gene CG5892? That would give a better sense of the orthology between the intergenic region that sometimes differs in size considerably between species and makes the task particularly risky. How were the CREs defined and how reliable are they in this context?

6) Another example of the necessity for more detailed methods relates to the results presented in Fig 3. It is unclear what is the reference against which relative intensity is being measured. When the authors claim that ‘Red and black arrowheads indicate low and increased levels of ebony mRNA’, is it relative to what? On the same Fig 3, it is not clear if panels H and I quantify pigmentation (panels B-D) or expression via in situ hybridization (panels E-G). Moreover, the H-I quantification misses segment A4, which appears to be most informative one, considering the adult’s final pigmentation phenotypes (panels B-D). This becomes more surprising if one considers that in Fig 2 the authors focus solely on that segment A4!

The last comment also relates to the fact that there seem to be some inconsistencies in the way the data is presented. There is for instance, a discrepancy between the segments being considered in each of the results section (sometimes it is A4 and A5, and sometimes A5 and A6), which makes hard to relate the results and weakens some of the conclusions drawn across experiments.

Another instance of inconsistency is the use of males and/or females. It is not always clear in each experiment, if one or the other, or both sexes have been considered. Please make this explicit in figure legends, and in the text be clearer on why a given option was made.

Minor Suggestions:

It would have been interesting to see the phenotypes against the ebony mutants in the in situ hybridization results (Fig. 3 and Fig S1). It is not a necessary experiment, but if that data would be available, it could be an interesting comparison to add.

Finally, the discussion could benefit from further developing the global versus local effects, particularly if the title includes the term ‘global repression’. Not excluding what was said before, maybe the title itself can be revised to a more general conclusion, for example: ‘The role of transcriptional repression in the evolution of Drosophila abdominal pigmentation.’

Other comments:

• Figure 1A: The authors could consider adding the names of the enhancer (eAct) and the silencers (eMS and eSS) to the schemes so that the figure matches better the text description in lines 37-44.

• Figures 1A, 2A, 2J, and 2K have very low resolution.

• Figure 1 legend, panel A: please define what the white and yellow boxes represent in the gene model and what are the red arrows pointing to.

• Figure 4, panel A: is hard to understand

• Figure 4, legend: add at the end of the last sentence ‘for D. ananassae, D. melanogaster, and D. pseudoobscura, respectively’.

• Line 90: D. melanogaster should be in itallics.

• Line 95: is missing a reference

• Lines 120-123: it is unclear if the pigmentation was measured in the entire abdomen or only in the anterior-most part.

• Line 145-147: It is unclear how the measurements described in this section relate to the squares drawn in Figure 4C-K. Also, in this same section, please confirm that the description of how the stripe silencing activity is measured is correct

• Line 148: please correct ‘A45 segment’.

• Line 159-160: the statement that ‘Deletion of eAct did not affect the abdominal pigmentation intensity (Fig 2D-D’, J-K)’ is confusing given that the corresponding images show no midline pigmentation.

• Line 188: wings are not shown in any figure but are referred to in the results

• Lines 201-202: please confirm that the figure panels cited in are correct.

• Lines 204-205: The proposed hypothesis of why there are no phenotypic effects observed (and what is the data behind it) could be better explained

• Line 209: delete extra space before ‘1.5 kb‘

• Line 201: please confirm that the cited figure panels are correct

• Line 236: remove ‘the’

• Line 242-243 : please confirm that the cited figure panels are correct

• Lines 245-247: it is unclear what is the rationale behind this hypothesis. Please explain

• Line 319: delete extra space

• Line 394: remove dot after ‘females’

**Have all data underlying the figures and results presented in the manuscript been provided?**

Reviewer #1: Yes

Reviewer #2: Yes

Reviewer #3: **No: **Should include accession numbers and/or sequences for all novel enhancers identified in this study, particularly for those without published genome assemblies (e.g., D. malerkoliiana).

Reviewer #4: Yes

PLOS authors have the option to publish the peer review history of their article (what does this mean?). If published, this will include your full peer review and any attached files.

Reviewer #1: No

Reviewer #2: No

Reviewer #3: No

Reviewer #4: No

---

## [Decision Letter · Decision Letter 1]

28 Mar 2023

Dear Dr Rebeiz,

We are pleased to inform you that your manuscript entitled "Changes in locus wide repression underlie the evolution of Drosophila abdominal pigmentation" has been editorially accepted for publication in PLOS Genetics. Congratulations!

Yours sincerely,

Artyom Kopp

Academic Editor

PLOS Genetics

Kirsten Bomblies

Section Editor

PLOS Genetics

Comments from the reviewers (if applicable):

Reviewer's Responses to Questions

**Comments to the Authors:**

Reviewer #2: The authors responded well to the suggestions of the reviewers, and I am happy with the paper as it currently stands.

Reviewer #3: The authors have done a fantastic job responding to all of my points in my first review with the changes they have made in the revised manuscript. The authors have also thoroughly explained their rationale for the changes in their response to reviewers. I am completely satisfied with the thoroughness of revisions. This is study is a solid addition to the canon of morphology-related gene regulatory evolution in Drosophila.

Reviewer #5: The authors have satisfactorily addressed our previous concerns. A few additional minor comments are listed below.  

Line 218: Add missing comma after ‘mRNA’.

Line 194: Add missing comma after ‘Importantly’.

Line 239: Add missing comma after references '(30,31)'.

Line 348: Remove extra space after ‘eAct’.

**Have all data underlying the figures and results presented in the manuscript been provided?**

Reviewer #2: Yes

Reviewer #3: Yes

Reviewer #5: Yes

PLOS authors have the option to publish the peer review history of their article (what does this mean?). If published, this will include your full peer review and any attached files.

Reviewer #2: No

Reviewer #3: No

Reviewer #5: No

**Data Deposition**

http://datadryad.org/submit?journalID=pgenetics&manu=PGENETICS-D-22-01315R1

**Press Queries**

---

## [Editor Report · Acceptance letter]

28 Apr 2023

PGENETICS-D-22-01315R1 

Changes in locus wide repression underlie the evolution of Drosophila abdominal pigmentation 

Dear Dr Rebeiz, 

We are pleased to inform you that your manuscript entitled "Changes in locus wide repression underlie the evolution of Drosophila abdominal pigmentation" has been formally accepted for publication in PLOS Genetics! Your manuscript is now with our production department and you will be notified of the publication date in due course.

With kind regards,

Anita Estes

PLOS Genetics

On behalf of:
